# Chrysoviruses Inhabited Symbiotic Fungi of Lichens

**DOI:** 10.3390/v11121120

**Published:** 2019-12-03

**Authors:** Karel Petrzik, Igor Koloniuk, Hana Sehadová, Tatiana Sarkisova

**Affiliations:** 1Department of Plant Virology, Institute of Plant Molecular Biology, Biology Centre, Czech Academy of Sciences, Branišovská 31, 370 05 České Budějovice, Czech Republic; koloniuk@umbr.cas.cz (I.K.); sarkisova@umbr.cas.cz (T.S.); 2Institute of Entomology, Biology Centre, Czech Academy of Sciences, Branišovská 31, 370 05 České Budějovice, Czech Republic; sehadova@entu.cas.cz; 3Faculty of Science, University of South Bohemia, Branišovská 31, 370 05 České Budějovice, Czech Republic

**Keywords:** saxicolous lichen, ascomycete, chrysovirus, complete genome, confocal microscopy

## Abstract

A lichen body is formed most often from green alga cells trapped in a net of ascomycetous fungi and accompanied by endolichenic or parasitic fungi, other algae, and symbiotic or free-living bacteria. The lichen’s microcosmos is inhabited by mites, insects, and other animals for which the lichen is a source of food or a place to live. Novel, four-segmented dsRNA viruses were detected in saxicolous *Chrysothrix*
*chlorina* and *Lepraria incana* lichens. Comparison of encoded genome proteins revealed classification of the viruses to the genus *Alphachrysovirus* and a relationship to chrysoviruses from filamentous ascomycetous fungi. We propose the names Chrysothrix chrysovirus 1 (CcCV1) and Lepraria chrysovirus 1 (LiCV1) as acronyms for these viruses. Surprisingly, observation of *Chrysothrix chlorina* hybridization with fluorescent-labelled virus probe by confocal microscope revealed that the CcCV1 virus is not present in the lichen body-forming fungus but in accompanying endolichenic *Penicillium*
*citreosulfuratum* fungus. These are the first descriptions of mycoviruses from a lichen environment.

## 1. Introduction

Fungi are ancient organisms whose major lineages originated in the Precambrian [1]. They are essential constituents of modern terrestrial and aquatic ecosystems and partners with plants and other organisms. Lichens also comprise ancient symbiotic associations, most often between ascomycetous fungi (rarely basidiomycetous) from classes Lecanoromycetes or Lichinomycetes and green algae or cyanobacteria [2]. This symbiosis is one of the most successful life forms in nature and allows the partners to expand into habitats where separately they would be rare or nonexistent (including extremely cold, hot, dry, or toxic environments). In addition to the two main partners, there are other lichenicolous fungi, endolichenic fungi, and lichen-associated bacteria within lichen thalli composing specific complex microbial microcosms [3,4,5]. Bacteria that live externally from the fungal cells or as endosymbionts contribute to a complex symbiotic network with multiple functions. They could complement the nitrogen budget and/or provide defense against lichen pathogens and feeders. They have lytic activities; produce bioactive substances, hormones, and antibiotics; metabolize decaying lichen material; and are present in the lichen–substrate interface [4]. On the other hand, it seems that growth of the bacteria is under control of the lichens. Lichens are only rarely eradicated by pathogens. It is assumed that growth of the parasites requires tolerance to lichen compounds or prior breakdown of the lichen’s chemical defense [2,6]. Some exceptions do exist, as in the case of various *Fusarium* species, and these parasites attack a variety of lichens and are tolerant of many lichen defense compounds [7]. Moreover, several species of *Caloplaca* are known to have parasitic phases in their life cycles during which they take over other crustose lichens [8,9]. A killer is the basidiomycete *Athelia arachnoidea*, which has been shown to attack and destroy *Lecanora conizaeoides* lichen [10]. Neither can a presence of specific viruses in lichens be excluded, because a wide range of mycoviruses and viruses specific to green algae and cyanobacteria are known to infect free-living ascomycetous fungi, algae, and cyanobacteria [11,12,13]. In addition, with the advent of next-generation sequencing technology, the strict borders previously assumed to exist between the host preferences of plant viruses, mycoviruses, insect viruses, human viruses, and the like have become less strict as interkingdom infections have been described. Cytorhabdovirus sequences similar to plant cytorhabdoviruses have been amplified from *Cladonia arbuscula* lichen, as has been an Apple mosaic virus (a plant ilarvirus) from different *Usnea* sp., *Xanthoria parietina*, and *Cladonia arbuscula* samples [14]. Conversely, some viruses similar to mycoviruses have been found also in plants [15,16], and chlorovirus ATCV-1 infecting eukaryotic green algae has been found to be a part of the human virome [17].

The lichen genus *Chrysothrix* Mont. is characterized by the yellow, lemon to bright yellow, or golden yellow thallus containing pulvinic acid derivatives that give the lichens their characteristic color. *Chrysothrix chlorina* (Ach.) J. R. Laundon (syn. *Crocynia chlorina*, *Lepraria chlorina,* gold dust lichens, sulfur dust lichens) is a morphologically simple lichen-forming ascomycetous fungus in the family *Chrysothricaceae,* class *Lecanoromycetes*. The lichen is a widely distributed saxicolous (growing on rocks) and crustose (strongly adhered to the substrate) organism from the lowlands to the mountains in the temperate and arctic zones of both the Northern and Southern hemispheres. It has a thallus without cortex and reproduces asexually. The lichen never has been found with ascomata or conidiomata [18]. There is speculation that the pulvinic acid derivatives protect the lichen from some herbivores and have an antibacterial activity against gram-positive bacteria [19]. Vulpinic acid from this lichen has emerged recently as a potential drug candidate in the therapy of atherosclerosis [20]. *Lepraria incana* (L.) Ach. (family *Stereocaulaceae,* class *Lecanoromycetes*) is a morphologically simple, granular, grayish or greenish lichen growing all over the world on bark, acidic rock, wood and soil in shady places [21]. This lichen is sterile and never develops fruiting bodies (ascomata or conidiomata).

Mycoviruses are widely distributed cohabitants in fungi. Recently, they have been found in Basidiomycota as well as in Ascomycota. Furthermore, Plasmodiophorids and Chytrids (Chytridiomycota) are known vectors of some plant viruses belonging to the *Bymo*-, *Beny*-, *Furo*-, *Peclu*-, and *Pomovirus* genera [22]. Only a small number of known mycoviruses have deleterious effects on their hosts. Most infections are asymptomatic and do not affect host vigor, growth rate, or multiplication. The genomes of known mycoviruses consist of ssDNA, ssRNA, and, more predominantly, dsRNA. The low-cost and simple protocol of Morris and Dodds [23] is widely used for extraction of dsRNA, which is indicative for a presence of replicative ssRNA viruses as well as for dsRNA viruses in hosts. This approach has allowed us to screen different lichen samples for viral nucleic acids. In this paper, we describe for the first time two novel chrysoviruses found in simple lichens and present their localization in the lichen body.

## 2. Materials and Methods

### 2.1. Lichen Samples

*Chrysothrix chlorina* sample ZSH was scraped from granite rocks in the Vltava River valley (49.0737153N, 14.4529036E) in the České Budějovice countryside (Czech Republic). The *Lepraria incana* lichen sample DK10 was from growths in crevices of shadowed, wet rock at a location about 20 km upriver from the first (48.8904328N, 14.3561922E). Material of non-lichen origin was manually removed before further manipulation. Samples were washed in distilled water, surface-sterilized in 5% sodium hypochlorite for 15 min, and washed twice in water. Due to a limited amount of the DK10 sample, only partial sequencing was performed with this lichen. All other experiments were performed with the ZSH sample.

### 2.2. Double-Stranded RNA Extraction and Virus Sequencing

Double-stranded RNA was extracted from about 0.1 g of wet lichen thalli by the CF-11 cellulose chromatography method, as described previously [23]. Extracted nucleic acid was diluted in 50 µl of water. Co-purified DNA and ssRNA were then removed by DNase I and S1 nuclease treatment for 15 min each. After phenol/chloroform extraction and ethanol precipitation, the dsRNA was diluted in 20 µl of TE buffer (10 mM Tris-HCl, pH 8.0, 5 mM EDTA) and separated on agarose gel. Total lichen RNA was isolated using a NucleoSpin RNA Plant Kit (Macherey Nagel, Düren, Germany) according to the manufacturer’s instructions.

Complementary DNA (cDNA) was prepared with Superscript III reverse transcriptase (Invitrogen, Carlsbad, CA, USA) and tagged random primer 5′-CGATCGATCATGATGCAATGCNNNNNN-3′. The random cDNA products were then amplified using a single specific primer, 5′-CGATCGATCATGATGCAATGC-3′ [24]. The pool of polymerase chain reaction (PCR) products was ligated to pGEM-T Easy vector (Promega, Madison, WI, USA) and used for transformation of NEB 10-beta competent *Escherichia coli* cells. The 5′ and 3′ terminal sequences of each viral genome segment were obtained using 5′/3′ rapid amplification of cDNA ends (RACE) protocol with specific primers (see Appendix A). The 18S rDNA sequence of lichen fungus was amplified with ITS1 and ITS4 primers [25], cloned, and then sequenced.

### 2.3. Lichen Symbionts Separation

Lichen alga isolation was prepared according to Gasulla et al. [26] from 20 mg of lichen thalli by one-step centrifugation through Percoll™. Briefly, about 20 mg of lichen thalli was surface-sterilized in 2% sodium hypochlorite, washed with sterile water, then homogenized in 0.3 M sorbitol in 50 mM HEPES, pH 7.5. After filtration, the filtrate was centrifuged at 500× *g* for 5 min. The pellet was resuspended in sorbitol buffer and loaded onto 80% Percoll™ in sorbitol buffer. After centrifugation at 10,000× *g* for 10 min, a green layer near the top was collected, resuspended in the buffer, and a second round of centrifugation was performed on 80% Percoll™. Collected green layer was diluted in sterile water and centrifuged at 1000× *g* for 10 min. The pellet was then resuspended in 2 mL of sterile water, sonicated for 30 s, then centrifuged at 500× *g* for 5 min. This treatment was repeated 5 times.

Total nucleic acid was prepared from the final pellet containing the algal cells and, after the first centrifugation, from the sediment containing both algal and fungal cells using a NucleoSpin RNA Plant Kit (Macherey Nagel) according to the manufacturer’s instructions.

Multiple sequence alignments were carried out using the CLUSTALx [27] and MEGA v.7 programs [28]. Maximum likelihood analysis was used to infer virus phylogeny with 1000 bootstrap replicates.

Relative concentration of the virus in lichen thalli was estimated from quantitative real-time PCR (q-RT-PCR) performed in the CFX96 Real-Time System (Bio-Rad Laboratories, Hercules, CA, USA) with primers 1901 and 1902 (see Appendix A) amplifying the GAPDH housekeeping gene [29] and 846 and 847 primers specific for the RNA4 segment of the virus.

### 2.4. In situ Hybridization

For in situ hybridization, Cy3-labeled probes MY1574 5′-TCCTCGTTGAAGAGC-3′ specific for a wide range of Eumycota [30] and 859 probe 5′-GGGCAAATAGAGAGAAGG-3′ for the dsRNA3 segment of CcCV1 were synthesized (Sigma–Aldrich, St. Louis, MO, USA). Standard techniques were used for thalli tissue paraformaldehyde fixation, dehydration, embedding in Paraplast^®^, sectioning to 6 µm, deparaffinization, and rehydration. *Penicillium citreosulfuratum* samples for hybridization were taken from aerial hyphae of the fungus cultivated 10 days on potato dextrose agar (PDA) plates. The samples were then treated with chitinase for 15 min in 1× phosphate-buffered saline buffer with 1% sodium dodecyl sulfate (SDS), pH 5.5, at 30 °C and then rinsed with distilled water. Prehybridization was performed in the hybridization buffer (0.9 M NaCl, 20 mM Tris-HCl (pH 7.2), 0.03% SDS and 20% formamide) at 46 °C for 3 h. The samples were then hybridized with the probes in concentration 5 ng per µl of hybridization buffer at 46 °C overnight. They were then rinsed with 20 mM Tris-HCl (pH 7.2), 0.01% SDS, and 250 mM, 88 mM, 62 mM, and 31 mM NaCl buffer at 46 °C for 20 min each. Some samples were stained with calcofluor white (Merck, Darmstadt, Germany) for 5 min. Stained sections were rinsed in distilled water, dehydrated, mounted in DPX mounting medium (Fluka, part of Fisher Scientific, Hampton, NH, USA), then viewed and imaged under a Fluoview FV3000 confocal laser scanning microscope (Olympus, Shinjuku, Tokyo, Japan).

## 3. Results

### 3.1. Lichen Identification

Based on morphological features, the lichen sample ZSH was identified as *Chrysothrix chlorina*. The sequence of the algal symbiont (MN396489) was 98% identical to the GU017647 sequence of *Asterochloris phycobiontica*. Partial ribosomal sequence of the lichen fungus was amplified with ITS1 and ITS4 primers, sequenced, then compared in BLAST with the GenBank database. The lichen of the DK10 sample was identified as *Lepraria incana*. The 18S rDNA sequence (MN393971) was 95% identical with *L. incana* sequence AF517899 deposited in GenBank. The algal symbiont of this virus (MN396484) was 97% identical to uncultured *Trebouxia* sp. sequence AY250848.

### 3.2. dsRNA Presence

In ZSH, a high molecular weight dsRNA band around 3.5 kbp was visible after separation on agarose gel (Figure 1). After sequencing, four components were recognized in this material. In DK10, four bands within the range 2 to almost 4 kbp were visible (Appendix A).

### 3.3. Concentration of the Virus in Lichen

We had expected the lichen-forming fungus to be the host of these extrachromosomal dsRNAs, because no high molecular weight dsRNA was visible after extraction from partially purified algal symbiont. Algal layer after the third round of sonication/centrifugation was used for RNA isolation and q-RT-PCR detection. Virus-specific detection primers were prepared according to the obtained sequences and RT-PCR detection was performed in parallel in algal symbiont as well as fungal symbiont. No signal increase was observed after 40 rounds of amplification.

Relative concentration of the virus in lichen thalli was estimated from total RNA isolations from three different samplings. In a single isolation, only the virus-specific signal in q-RT-PCR occurred 10 cycles after the GAPDH signal, while in the other two isolations the virus concentration was below the detection level. This means that the relative concentration of the virus in ZSH samples varies significantly and was as little as one-thousandth the concentration of GAPDH.

### 3.4. Genome Description of the Viruses

We determined the complete nucleotide sequence of all dsRNA segments in the ZSH sample. The 5′- and 3′-termini of the four dsRNAs share conserved sequences (Figure 1). The (CAA)_n_ repeat that works as the enhancer element in chrysoviruses was present 5 to 9 times in the 5′-untranslated regions (UTR) of distinct segments. Only for the DK10 isolate were 5′-terminal regions of dsRNA1 and dsRNA2 segments obtained. Both revealed significant nucleotide sequence identity in that region, and this supports the assumption that they belong to a single virus.

The dsRNA1 segment from the ZSH sample was 3552 nt long and contained a single open reading frame (ORF) of 1089 amino acids (aa) with predicted molecular mass 125.5 kDa. The eight conserved RNA-dependent RNA polymerase motifs (RdRP_4) characteristic of RNA mycoviruses were localized in the central part of the protein (Figure 2). The polymerase of the ZSH isolate shows significant similarities to its counterpart in Penicillium roseopurpureum chrysovirus 1 (61% identity, Figure 3). The dsRNA2 segment was 3191 nt long and encoded a protein of 947 aa (predicted molecular mass 107 kDa) showing highest similarity to the capsid protein (CP) of Penicillium raistrickii chrysovirus 1 (36%) (Figure 4). The dsRNA3 segment was 2879 nt long and encoded a single protein of 848 aa (predicted molecular mass 97 kDa). This protein is only distantly related to the corresponding protein of other chrysoviruses. The corresponding AfuCV protein was recognized as the most similar (24.3% aa identity) (Appendix A). The dsRNA4 segment was 2776 nt long and encoded a protein of 840 aa (predicted molecular mass 94 kDa). Presence of ubiquitin thioesterase domain (E = 0.000011) was predicted in the C-terminal part of the protein by HHpred [31]. A hypothetical protein of AfuCV was identified as the most similar, with 45% aa identity (Appendix A).

Partial sequences of dsRNA1, dsRNA2, and dsRNA4 segments of a chrysovirus were obtained from the DK10 sample. We did not obtain a sequence of the fourth segment visible on the gel. These sequences represent 84%, 78%, and 72% of segments size for the most similar Colletotrichum gloeosporioides chrysovirus 1 (Appendix A) and reveal 51%, 36%, and 37% identity with the corresponding proteins of CgCV1. All eight conserved RdRp motifs characteristic of dsRNA mycoviruses were detected on the in silico translated sequence of the dsRNA1 segment and shared high identity with related alphachrysoviruses (Figure 2).

### 3.5. Taxonomic Relationship

The taxonomic criteria for differentiating species within the family *Chrysoviridae* reflect host isolation, less than 70% aa sequence identity in the RdRp, size of segments, length of 5′-UTR, and serological relationships [32]. In the phylogenetic tree computed on RdRp aa sequences, the virus from the ZSH sample was placed close to Isaria javanica chrysovirus 1 (Figure 5). The virus from the DK10 sample was placed in a branch together with Colletotrichum gloeosporioides chrysovirus. Similar relationships were observed also on the phylogenetic tree computed from the CP aa sequences (Figure 6). Based on the RdRp aa sequence identities and with respect to the other taxonomic criteria for chrysoviruses, we concluded that the two viruses detected in the lichen samples represent novel species in the genus *Alphachrysovirus*. We propose for these viruses the names Chrysothrix chrysovirus 1 (CcCV1) and Lepraria chrysovirus 1 (LiCV1), reflecting the names of their host sources.

### 3.6. Virus Localization in Lichen

Universal fungus-specific probe and CcCV1 RNA3 segment probe were used to localize the virus inside the lichen components. In the conditions used for the hybridization (46 °C, 20% formamide, overnight hybridization), the fungus probe penetrated hyphae and gave strong signal in the majority of hyphae (Figure 7A). The accompanying algae cells did not hybridize with the probe. This signal correlated with calcofluor white (Merck) staining of chitin. The CcCV1 RNA3 segment-specific probe Cy3-859 hybridized with material present in the cytoplasm of only a small proportion of hyphae (Figure 7B,C). These hyphae showed denser cytoplasm content and differed from the hyphae forming the lichen thallus.

We hypothesized that the hyphae positive for the virus presence are not hyphae of the lichen body-forming *Chrysothrix* but of some accompanying or endolichenic fungus. To resolve this question, we cultivated fungus-enriched fraction on PDA plates for 10 days. Newly growing fungi were picked and cultivated individually on fresh PDA plates and used for dsRNA extraction, RT-PCR virus detection, and hybridization. *Penicillium citreosulfuratum* from the *P. citreonigrum* clade [33] was identified as the host of CcCV1, inasmuch as RT-PCR with virus-specific primers gave products of expected size (Figure 8B) and the fungus reacted with the Cy3-859 probe (Figure 8D–G). The transmission of CcCV1 is uncertain. We could expect that persistence and transmission of CcCV1 would be tightly joined with the life strategy of this endolichenic fungus. In additional testing, RT-PCR screening detected CcCV1 in all single spore cultures of *P. citreosulfuratum*. In the hybridization experiments with the virus probe, strong signal is observed from conidiophores and from single spores of *P. citreosulfuratum*. That is consistent with the fact that the virus is frequently transmitted by conidia (Figure 8D-G). To highlight the lichen *Chrysothrix* as the complex sample, however, where the virus was first identified and where the *P. citreosulfuratum* is only a minority cohabitant, we propose the virus name be assigned according to the lichen.

## 4. Discussion

Chrysoviruses and their related viruses comprise three to five linear monocistronic dsRNA segments encapsidated separately in identical capsids 35–40 nm in diameter [34]. Most often, ascomycetous fungi are hosts of distinct chrysoviruses, but Agaricus bisporus virus 1 lives in a basidiomycete; RsCV1, BcCV1, Persea americana chrysovirus, ACDACV, and Anthurium mosaic-associated virus infect plants; one chrysovirus sequence was obtained from fruit fly *Drosophila melanogaster* [35]; and recently, Hubei chryso-like virus and Shuangao chryso-like virus were discovered in *Culex* mosquitos [36]. Furthermore, chrysovirus-like sequences have been identified in cDNA libraries of plant genomes, where they possibly represent the co-evolved viral lineage in plants [37]. The overall size of the genome of chrysoviruses ranges from 8.9 to 16.0 kbp. This concept of separately encapsidated dsRNA genomic monocistronic segments is probably highly effective in the evolution of mycoviruses, where viruses in the family *Partitiviridae* with two essential genome segments are the mycoviruses most often found (about 200 species) [38]. Viruses in the family *Chrysoviridae*, with 3–5 segments (25 classified species and about 20 chryso-like viruses), make up the second [39]. Mycoviruses with 4 and 11 dsRNA segments are known and classified in the family *Quadriviridae* and genus *Mycoreovirus* (one species each), respectively.

The transmission of chrysoviruses and especially CcCV1 is uncertain. Mycoviruses themselves have limited ways of transmission and, with several exceptions, they generally do not have extracellular routes for infection and the viral particles are not infectious. Usually they are transmitted via hyphal anastomosis and heterokaryosis or via sexual and asexual spores. In chrysoviruses, hyphal fusion experiments have demonstrated that MoCV1 is transmissible via anastomosis [40]. Transmission through ascospores observed in AthCV1 was relatively low at 37%. The infectivity of AthCV1 particles has been documented by successful transfection of protoplasts [41]. Based on our hybridization experiments, CcCV1 is highly transmissible by conidia of *P. citreosulfuratum.*

The novel mycoviruses in *Chrysothrix*- and *Lepraria*-lichenized fungi are related to mycoviruses infecting filamentous fungi and could be classified as regular species of the genus *Alphachrysovirus*. One can only speculate thus far about the significance of these viruses in lichens, but, analogously with viruses of higher plants and fungi, we could expect an influence on lichen viability, stress tolerance, changes in morphology and/or in gene expression, and other effects. This influence could be directly on the lichen or on the accompanying endogenous fungus. In any case, the cohabitation of the *Chrysothrix* lichen, *Penicillium citreosulfuratum* endolichenic fungus and CcCV1 (Figure 9) is persistent in time, as the first samples were collected from the locality 5 years ago and CcCV1 has been detected there repeatedly up to the present. Its concentration in lichen samples as measured by extracted dsRNA varied, however, probably in relation to variable colonization by the endolichenic fungus. Concentration of CcCV1 in pure laboratory cultures of *P. citreosulfuratum* remains stable at a high concentration that is about one-tenth that of the GAPDH reference gene (Figure 8C). Furthermore, the lichens could serve as reservoirs for mycoviruses as well as for herbaceous viruses, despite that the manner of transmission between different organisms is not clear [14].

Although most of the reported mycoviruses have been associated with cryptic or latent infections of their hosts [42], chrysoviruses relatively often have been associated with the hypovirulence phenomenon. For example, FodV1 has been associated with the induction of hypovirulence in *Fusarium oxysporum* f. sp. dianthi, where the virus-infected strain appeared in the intercellular spaces and with a lower colonization density in external roots [43]. Some tentative members of the family *Chrysoviridae*, such as Botryosphaeria dothidea chrysovirus 1 [44], Magnaporthe oryzae chrysovirus 1-A [45], Magnaporthe oryzae chrysovirus 1-B [46], and Agaricus bisporus virus 1 [47], are known to decrease virulence or cause other phenotypic changes in their fungal hosts. AaCV1 exhibits impaired growth of the host fungus and increased level of the host-specific AK-toxin [48]. The *Alternaria alternata* containing the AaV1 had an abnormal growth, reduced mycelial growth, aerial mycelial collapse, unregulated pigmentation and cytolysis, while the cured strain had a normal mycelial growth, restored pigmentation and virus content only about one-tenth as great [49]. A presence of Aspergillus thermomutatus chrysovirus 1 resulted in large ascospores production, but its host, even as the virus-free culture line, produces no ascospores. Moreover, the conidiation was 10 times greater in the virus-containing line compared to the virus-free line [41]. *Cryphonectria nitschkei* BS122 infected with Cryphonectria nitschkei chrysovirus 1 manifested reduced mycelial growth in comparison to the cured isogenic strain [50]. No such effects were observed in CcCV1-infected *P. citreosulfuratum*, however, and this cohabitation was similar to an infection by Isaria javanica chrysovirus or Raphanus sativus chrysovirus 1 in its hosts [39,51].

## 5. Conclusions

A new virus distantly related to Aspergillus fumigatus chrysovirus and named Chrysothrix lichen chrysovirus 1 (CcCV1) has been found to inhabit *Chrysothrix chlorina* lichen. The new virus was completely sequenced and analyzed. Confocal microscopy observation of the virus hybridization signal revealed that not the lichen fungus, but an endolichenic fungus *Penicillium citreosulfuratum* was the main host where CcCV1 replicated. A partial genome sequence of another novel chrysovirus named Lepraria lichen chrysovirus 1 (LiCV1) was obtained from *Lepraria incana* lichen. Both these viruses should be classified in the genus *Alphachrysovirus*.

## Figures and Tables

**Figure 1 viruses-11-01120-f001:**
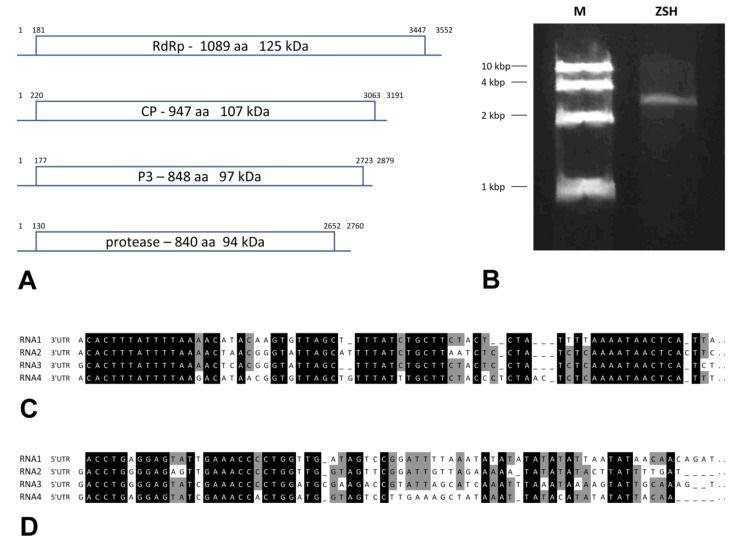
Genome characteristics of Chrysothrix chrysovirus 1 (CcCV1) from the ZSH sample. (**A**) Genome arrangement. (**B**) dsRNA isolated from ZSH lichen sample. M-DNA size standard of 10 kbp, 4 kbp, 2 kbp, and 1 kbp. (**C**) Alignment of 3′UTR (untranslated region) sequences of distinct segments of CcCV1. (**D**) Alignment of 5′UTR sequences of distinct segments of CcCV1.

**Figure 2 viruses-11-01120-f002:**
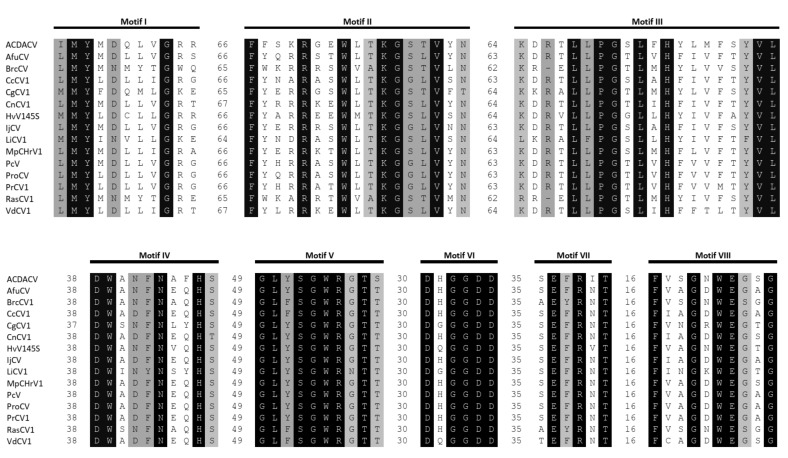
Alignment of amino acid sequences of the conserved I to VIII RdRp motifs of alphachrysoviruses. For virus acronyms, see Appendix A.

**Figure 3 viruses-11-01120-f003:**
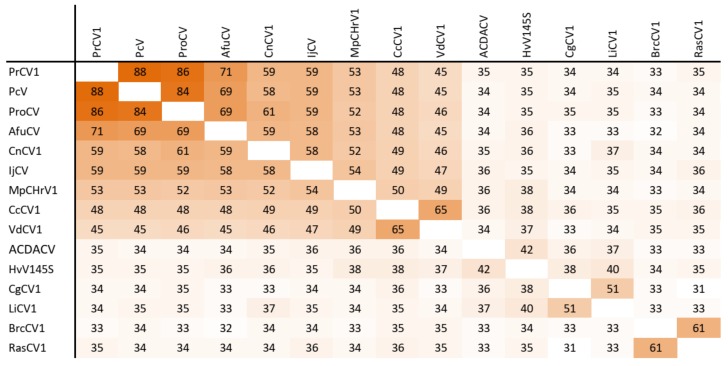
Amino acid sequence identity and heat map of RdRp protein of alphachrysoviruses. A darker color indicates a higher sequence identity.

**Figure 4 viruses-11-01120-f004:**
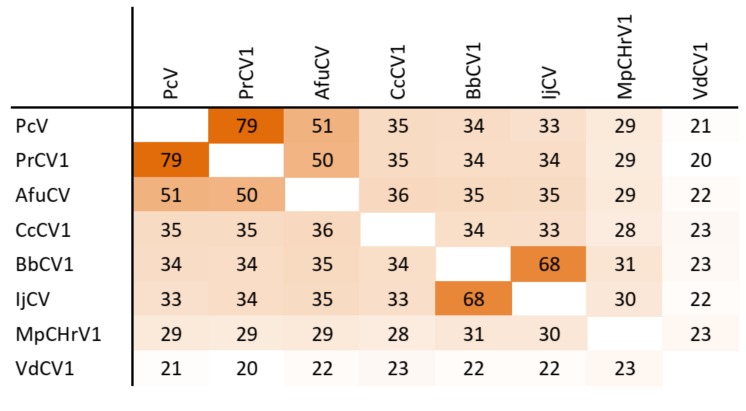
Amino acid sequence identity and heat map of capsid protein of chrysoviruses. A darker color indicates a higher sequence identity.

**Figure 5 viruses-11-01120-f005:**
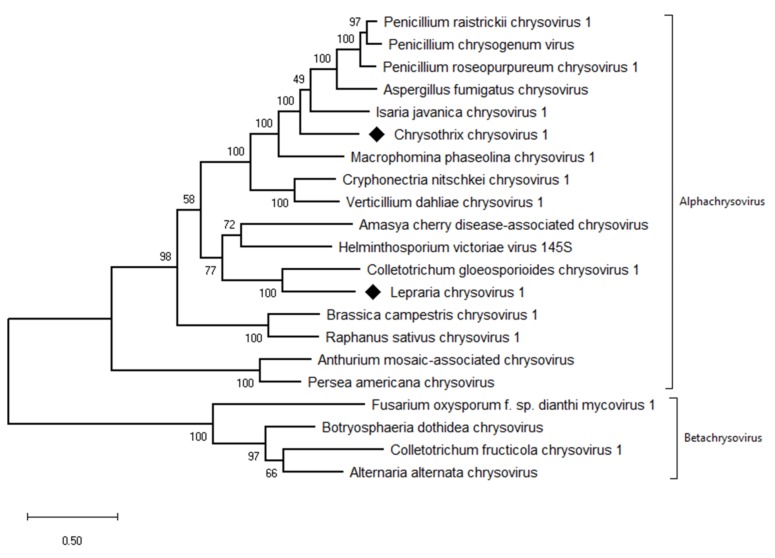
Phylogenetic maximum-likelihood tree computed on RdRp amino acid sequences of alphachrysoviruses and related betachrysoviruses. The values at the nodes are bootstrap values estimated by 1000 replicates. Diamonds highlight the viruses of this work.

**Figure 6 viruses-11-01120-f006:**
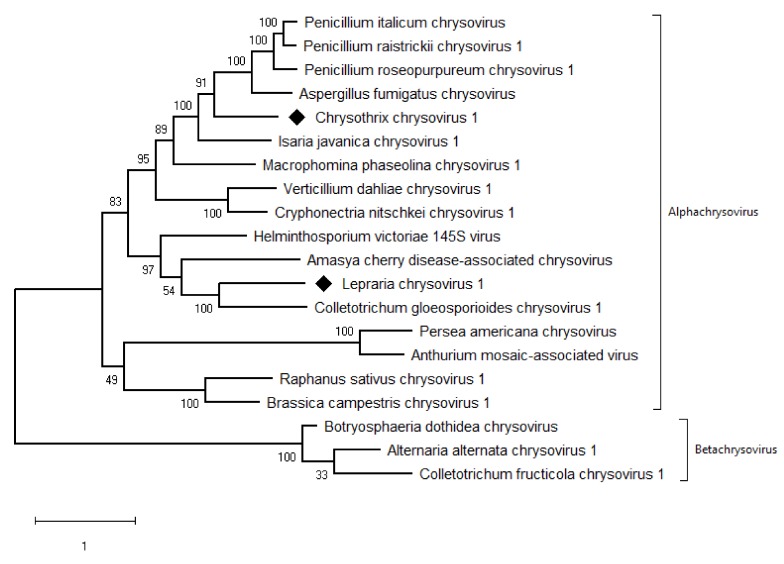
Phylogenetic maximum-likelihood tree computed on capsid protein amino acid sequences of alphachrysoviruses and related betachrysoviruses. The values at the nodes are bootstrap values estimated by 1000 replicates. Diamonds highlight the viruses of this work.

**Figure 7 viruses-11-01120-f007:**
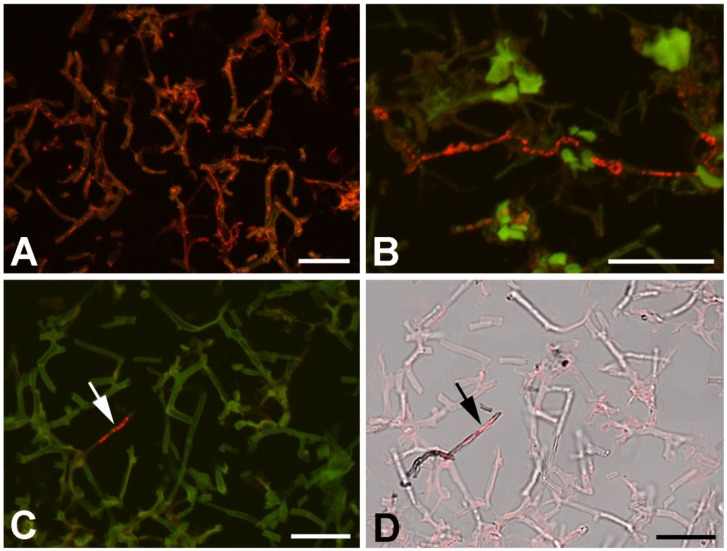
Confocal image of lichen thallus hybridized with fungus- and virus-specific probe. (**A**) Strong signal of lichen body-forming fungus with universal fungus-specific probe. (**B**,**D**) Cy3-859 hybridization signal specific for RNA3 segment of CcCV1 in distinct hyphae (in red). Positive hyphae show different color and structure (arrows). In **A**–**C**, the positive red signal is merged with an image of autofluorescence (green) excited by light of wavelength 488 nm. In **D**, the positive signal (red) is merged with a bright field image. Scale = 20 µm.

**Figure 8 viruses-11-01120-f008:**
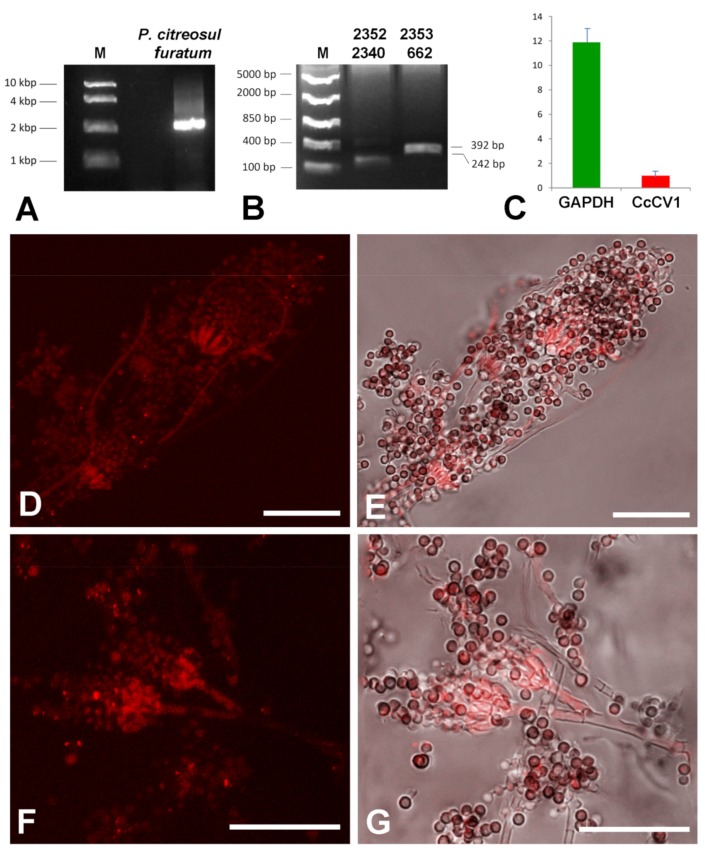
Characterization of CcCV1 in the endolichenic fungus. (**A**) dsRNA extracted from *Penicillium citreosulfuratum* endolichenic fungus, (**B**) PCR detection of CcCV1 with 2352 and 2340 (242 bp product), 2353 and 662 specific primers (392 bp product), (**C**) amount of CcCV1 relative to GAPDH housekeeping gene, (**D**–**G**) hybridization of pure culture of the endolichenic fungus with Cy3-859 probe. In (**E**,**G**) the positive signal (red) is merged with a bright field image. Scale = 20 µm.

**Figure 9 viruses-11-01120-f009:**
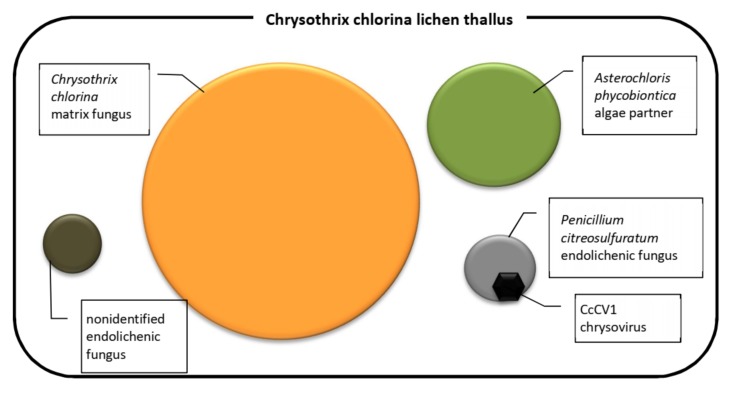
Components of the *Chrysothrix chlorina* lichen thallus.

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
