# Peer review of "Chrysoviruses Inhabited Symbiotic Fungi of Lichens"

_viruses, 2019, doi:10.3390/v11121120_

Round 1

Reviewer 1 Report

This research demonstrated that two chrysoviruses were identified and classified as regular species of the genus Alphachrysovirus.  The authors classified them to belong to chrysoviruses similar to those discovered from filamentous ascomycetous fungi. Moreover, they determined that the CcCV1 virus locates in the symbiotic fungi of lichen by hybridization with fluorescent-labelled virus probe. The study is interesting for further exploration of the function of these viruses in fungi as part of the lichens. Some of main conclusions were supported by data but some concerns were also noticed, which are listed below.

Major concerns:

1.       The presence of one virus (CcCV1) were detected in the endophytic fungus, but not the body-forming fungus. The reviewer is not sure about naming the virus with the body-forming fungus as the host for CcCV1.

2.     In situ hybridization only showed that the virus was located in P. citreosulfuratum, and this reviewer couldn’t conclude if P. citreosulfuratum is the only host based on the available data. Also, if P. citreosulfuratum is endosymbiont growing in the body-forming fungus, should not there viruses detected in Chrysothrix too?

3. What about the other virus (LiCV1)? Is it known whether LiCV1 actually present in the body-forming fungus?

4. In this day and era, metatranscriptome should have been used instead of dsRNA extraction. The cost is not an issue anymore. Then there won't be a missing information on an RNA segment for LiCV1.

5. Or protoplast preparation and virus reintroduction into virus-free isolate should be performed to conclude that P. citresosulfuratum is the host.

Minor concern:

1. In Fig.7B, C. images, the viruses were hybridization with the virus-specific probe and found located in the hyphae, which lead the authors concluded that the virus infect the cytoplasm. It seems the conclusion is weakly supported since the reviewer cannot distinguish the cytoplasm from nucleus. TEM is needed to distinguish between cytoplasm from nucleus where the virus replicates.

Author Response

Ad1) The rules for naming new viruses are described in the ICTV code (October 2018) https://talk.ictvonline.org/information/w/ictv-information/383/ictv-code. There is no rule against using complex host name like lichens are for naming the virus. On the other hand, the giant DNA virus infecting chlorella alga was named Paramecium bursaria chlorella virus 1 according to a complex chlorella algae host. As we know, there is no precedent for naming viruses from lichens. We believe that the information that lichens may contain viruses will be more important than Penicillium, which contains a virus.  

Ad2) From the beginning we used Chrysothrix as the original sample without detailed discrimination of the host. Later on we were able to separate the algal symbiont and cultivate it in an axenic culture, but not the fungal partner. Another fast growing fungi were cultivated from Chrysothrix lichen. P.citreosulfuratum was the only one species recognized as host of CcCV1 by dsRNA extraction, RT-PCR, and hybridization. 

Ad3) In case of the virus from Lepraria, the sample was collected almost 5 years ago and was consumed in dsRNA extractions. Three partial sequences were obtained. This  incomplete data was used in this MS to present another lichen host for chrysoviruses and to alert lichens as virologically unexplored group. The collection site (a rock near a building) has been cleaned in the meantime and no longer exists. 

Ad4) Of course. This approach has been used for CcCV1.

Ad5) Twenty monosporic cultures of P.citreosulfuratum was prepared and RT-PCR tested positive for CcCV1. More probably, non-host fungus will not transmit the virus by spores so effectively. We believe that together with almost ubiquitous probe signal in (not on the surface) P.citreosulfuratum hyphae, conidiophores, and conidia, this is good proof for discrimination of P.citreosulfuratum as a host of CcCV1.

Final: It is a question where the chrysoviruses replicate as this process has not been characterized in detail. Accumulation of virions in the cytoplasm is mentioned in Chrysovirus characteristics in (DOI 10.1099/jgv.0.000994). The spread and accumulation of the virus in the germinating hyphae will be the subject of further research.

Reviewer 2 Report

  There are many editorial mistakes in the current version of the manuscript. I hope the authors carefully read the mansucript and correct the mistakes. For example,

Line 2, In the tiltle, "in" should be added before "symbiotic" Line 12, "A" should be added before "lichen body" line 13, change "fungus" to "fungi" line 17, write "Alphachrysovirus" in italic line 31, change "cyanobacterium" to "cyanobacteria" Line 43, Write "Fusarium" in italic Line 55, "some mycoviruses-associaed viruses" not clear in meaning

   The other thing is that the authors should add a few subtitles in "Materials and Methods"

Author Response

The MS has been thoroughly read by English native speaker and the errors were corrected as well as the typography of latin names and virus taxons.

An English native speaker confirmation is attached.

Round 2

Reviewer 1 Report

N/A